# Neuroimaging Approach: Effects of Hot and Cold Germinated Wheat Beverages on Electroencephalographic (EEG) Activity of the Human Brain

**DOI:** 10.3390/foods12183493

**Published:** 2023-09-20

**Authors:** Thinzar Aung, Bo Ram Kim, Han Sub Kwak, Mi Jeong Kim

**Affiliations:** 1Department of Food and Nutrition, Changwon National University, Changwon 51140, Republic of Korea; thinzar@changwon.ac.kr; 2Interdisciplinary Program in Senior Human Ecology, Changwon National University, Changwon 51140, Republic of Korea; kinj56@changwon.ac.kr; 3Research Group of Food Processing, Korea Food Research Institute, Wanju-gun 55356, Republic of Korea; hskwak@kfri.re.kr

**Keywords:** electroencephalography, germinated wheat beverage, brain waves, consumption

## Abstract

Neuroimaging studies using electroencephalography (EEG) have been crucial in uncovering brain activity in sensory perception, emotion regulation, and decision-making. Despite tea’s global popularity, its temperature-related neural basis remains underexplored. This study investigated the effect of hot and cold germinated wheat beverages (HB and CB) in changes of brain waves using EEG. Four distinct approaches and topographical assessments were performed to gain deeper insights into the impact of EEG signals in the human brain. The four approaches showed different impacts of HB and CB intake, as all EEG spectral powers increased after drinking HB and decreased after consumption of CB. Significant increases in delta and theta waves were observed as a result of drinking HB, but significant decreases in alpha and beta waves were observed after drinking CB. The topographic maps illustrate the significant effects of HB more prominently than those of CB, displaying greater changes in delta, theta, and beta. These findings suggest the intake of HB is probably related to relaxation, calmness, mindfulness and concentration, while the intake of CB is related to alertness, attention, and working memory. Ultimately, the neuroscientific approaches provided in this study could advance consumer-based research on beverage consumption.

## 1. Introduction

A beverage prepared from roasted germinated wheat (*Triticum aestivum* L.) is a recently developed functional beverage [1,2,3]. Previous reports have described hot and cold germinated wheat beverages containing abundant volatile compounds, amino acids, and phenolic compounds. In line with our previous report, cold beverages were found to have a significant abundance of amino acids, contributing to sourness and umami taste traits. Conversely, hot beverages exhibited higher levels of volatile compounds, particularly hydrocarbons, which contributed to their pronounced odor profiles [2]. To obtain a marketable product, the sensory profiles and intensities of these beverages were measured using electronic sensors [2,4]. In addition to consumers’ sensory and emotional responses, their physiological responses could be an interesting topic to explore in sensory research [4]. When consumers eat or encounter food aromas, their food choices are typically influenced by two types of responses: physiological responses such as brain activity, autonomic nervous system activity, and facial expression, and emotional responses including happiness, like/dislike, and neural emotion [5]. Since it is difficult to understand the processes of the brain, studying brain activity using neuroimaging has been suggested as an advanced tool to understand the signaling of brain dynamics [6]. Among neuroimaging techniques, electroencephalography (EEG) is particularly applied in consumer sensory science to monitor brain waves in the form of alpha, beta, theta, and delta waves after food consumption [5]. In the development of healthy food, the behavior of the brain upon food consumption, indicating a positive or negative effect on consumers, is important, and the EEG technique can provide an in-depth understanding of human brain function [5].

In an EEG study, power spectra are determined by fast Fourier-transform (FFT) analysis of the signals by recording the signals from the scalp or surface of the brain [7]. In human brain, there are five basic categories of brain waves: delta waves observed during sleep and dreaming; theta waves associated with states of deep relaxation, meditation, daydreaming, and light sleep; alpha waves typically related to relaxed but alert states, such as meditating or resting; beta waves associated with active thinking, concentration, alertness, and problem-solving; and gamma waves associated with heightened mental activity, perception, and learning [8]. Generally, EEG spectral bands applied for the analysis of brain activity are delta (1–4 Hz), theta (4–8 Hz), alpha (8–12 Hz), beta (12–25 Hz), high beta (25–30 Hz), gamma (30–40 Hz), and high gamma (40–50 Hz) [9]. These waves reflect neuronal electrical activity in the brain and can be affected by sleep, brain disorders, medication, and age [7]. Delta waves are the slowest waves, associated with deep sleeping conditions (dreamless sleep) [7]. Theta waves are presumed to occur during sleep and deep meditation when the brain is subconscious [7]. Alpha waves correlate with calmness and alertness, while beta waves indicate drowsiness or a highly alert state [7].

The quantification and interpretation of EEG signals are prerequisites for a better understanding of the results. Several statistical assessments are applied to achieve an insightful estimation of specific results, and there are pros and cons to using these techniques. The application of multiple perspectives in EEG analysis may be useful for diagnostic and prognostic purposes [10]. They implicate electrode-based, cortical-based, hemisphere-based, anterior-posterior-based, and total cortex-based analyses, as well as conversion into topographical illustrations describing the highlights and challenges of these techniques. In food-related brain wave analyses, asymmetry of EEG signals from the right and left hemispheres, anterior and posterior lobes, and frontal, parietal, and occipital cortical subregions are generally used to determine consumer physiological responses after food consumption [5]. Since the anterior areas are associated with cognitive performance and motivation, and the posterior areas are related to sensory and somatosensory responses, anterior-posterior analysis could provide an appropriate comparison for brain wave changes [10]. Hemisphere-based analysis (left and right hemisphere) of EEG signals reflects motivational tendency or food acceptability in the evaluation of consumer responses in sensory research [5,11]. Finally, a topographic map with the display of cortical activity is also employed as it provides a clear and accurate interpretation of EEG data with a color-coded head-shaped diagram representing the electrical signals from different brain regions at specific frequency bands [10].

The effects of hot and cold beverage consumption on EEG brain waves has been studied broadly. Both cold and hot beverages have been found to affect cognitive performance. Studies have shown that hot drinks can improve attention, memory, and cognitive processing speed, while cold drinks can improve reaction time and cognitive performance in tasks requiring sustained attention. In addition, hot beverages have been associated with relaxation and a sense of calm, leading to increased alpha brain wave activity, while cold beverages can lead to increased beta and gamma wave activity, which are associated with wakefulness and attentiveness. Thus, it is worth noting that individual variations and factors such as the specific beverage consumed, and its temperature or components, can influence the observed effects on brain wave patterns. However, there is a lack of research specifically focused on the impact of germinated wheat beverages prepared in hot and cold conditions. Therefore, the primary objective of this study was to investigate the impact of consuming hot and cold germinated wheat beverages on brain wave patterns, aiming to enhance our understanding of the neurophysiology of consumers. With this aim in sight, hot and cold germinated wheat beverages that had the best consumer responses in previous studies were selected to measure brain waves (delta, theta, alpha, and beta) before and after consumption of selected beverages. Regarding the effect of hot and cold beverages on changes in waves with specific cognitive performance, four different statistical approaches (cortex-based, anterior-posterior-based, hemisphere-based, and whole cortex analyses) were applied as a first attempt in germinated wheat beverage-related neuroimaging studies. Then, the colored topographic map was used to display significant changes over the whole cortex upon intake of hot and cold beverages. Our study provides insights into the differential changes in neural activity as a result of the consumption of hot and cold germinated wheat beverages. Furthermore, our study revealed that region-specific analyses of the cortex provide a deeper understanding of cognitive performance in neuroimaging research.

## 2. Materials and Methods

### 2.1. Samples

Beverage samples were prepared using the method described in our previous study [2]. We germinated raw wheat at 17.6 °C for 46.18 h before roasting the wheat at 180 °C for 45 min [1]. Then, roasted germinated wheat (RSGW) was prepared for hot beverages (HB) and cold beverages (CB) based on the previously optimized preparation methods [2]. HB was obtained by infusing a tea bag filled with 0.8 g of RSGW in boiling water (100 mL) for 25 min, and CB was collected by boiling 20 g RSGW in water (1000 mL) for 30 min, followed by cooling at 4 °C. During the EEG test, participants were served 100 mL of HB after a 2 min wait at 60 ± 2 °C, while 100 mL of CB was served after being removed from the refrigerated environment at 4 °C. Both were freshly prepared, and EEG analysis was performed before and after the consumption of each beverage.

### 2.2. Participants

The EEG measurements were performed at the sensory laboratory of the Korea Food Research Institute. Thirteen participants (9 males and 4 females) between 21 and 28 years of age were recruited from Changwon National University, Changwon, Republic of Korea and the Korea Food Research Institute, Wanju-gun, Republic of Korea to participate in this experiment with the approval of the Changwon National University Institutional Review Board (IRB No:7001066-202002-HR-004). All participants were healthy and provided informed consent for voluntary participation prior to the start of the experiment.

### 2.3. Electroencephalography (EEG) Data Acquisition

EEG raw data were recorded using a 256-channel high-density EEG system (Geodesic EEG System 400, Magstim EGI, Eugene, OR, USA) with high-density sensor (Geodesic Sensor Net 200, Magstim EGI, Eugene, OR, USA) electrodes. Electrodes were positioned on the scalp using 10–20 international standards, as follows: prefrontal pole (FP1, FP2), frontal (F3, F4), central (C3, C4), parietal (P3, P4), occipital (O1, O2), temporal (F7, F8), temporal center (T3, T4), temporal posterior (T5, T6), midline frontal (Fz), midline central (Cz), and midline parietal (Pz) (Figure 1a,b). After attaching the EEG electrodes to the head, the study participants settled and EEG spectral power was measured for 4 min; the participants then consumed the presented germinated wheat beverage, settled for 2 min, and EEG spectral power was measured for 4 min. During EEG recording, hot and cold beverages were alternately presented in the order of hot–cold and cold–hot to acquire more reliable and accurate data. Brain waves were determined for delta (1–4 Hz), theta (4–8 Hz), alpha (8–12 Hz), beta (12–25 Hz), high beta (25–30 Hz), gamma (30–40 Hz), and high gamma (40–50 Hz) using FFT [9].

### 2.4. Data Processing and Statistical Analysis

The detected EEG signals were analyzed using the NeuroGuide program (Applied Neuroscience, Inc., Largo, FL, USA). A normal section of the entire EEG signal was selected for approximately 10 s, and the EEG signal of the section to be analyzed was selected using an automatic selection function. Raw data were edited to screen physical artifacts (drowsiness, pupil movement, and muscle movement) by selecting ‘high’ sensitivity. The detection of artifacts was performed by utilizing Z-scores to compare them against a predetermined threshold. The threshold for Z-score was set at 2.00. Points exceeding the threshold were flagged as potential artifacts and subsequently removed using artifact rejection technique. The edited EEG data were selected for at least 40 s of EEG segments free of artifacts for further data analysis. FFT absolute power (μV^2^) was automatically calculated using the NeuroGuide software (v 3.1.0), displaying the threshold for Z-scores [12]. The measured EEG of each participant was compared using the Neurostat program (Applied Neuroscience, Inc., Largo, FL, USA) before and after drinking germinated wheat beverages using a *t*-test.

To achieve a clear comparison, the spectral power of each wave (delta, theta, alpha, and beta) at different electrode positions before and after drinking HB and CB was characterized at four different focuses, as described by Minnerly et al. (2021) using a paired *t*-test [10]. Cortex-based analysis was conducted by comparing the mean values of six groups: prefrontal (FP1 and FP2), frontal (F3, F4, F7, F8, and Fz), central (C3, C4, Cz, T3, and T4), temporal (T5 and T6), parietal (P3, P4, and Pz), and occipital (O1 and O2). Anterior-posterior analysis assessed signals from two subregions of the cortex: anterior (FP1, FP2, F4, Fz, F7, and F8) and posterior (O1, O2, P3, P4, Pz, T5, and T6), excluding the central regions (T3, T4, Fz, Cz, and Pz). Hemisphere-based analysis was performed as another approach, in which signals from the left hemisphere (FP1, F3, F7, C3, T3, T5, P3, and O1) and right hemisphere (FP2, F4, C4, T4, T6, P4, and O2) were investigated while eliminating the central subregions (Fz, Cz, and Pz). Total cortex analysis was conducted to analyze the spectral power differences across the entire cortex using signal data from all electrodes. Statistical analysis was performed to obtain mean values ± standard error of the mean (SEM) using XLSTAT software (version 2021.2, Addinsoft, Paris, France) and SPSS (v.27; IBM Corp, Armonk, NY, USA). A two-tailed test (*t*-test) was used to determine significant differences (*p* < 0.05).

## 3. Results

### 3.1. Cortex-Based Analysis

The changes in EEG spectra at six cortical subregions (prefrontal, frontal, central, temporal, parietal, and occipital) were grouped and analyzed by using a paired *t*-test, showing the significance of differences between before and after drinking HB and CB (Figure 2). Although elevated spectral power was observed for delta in all cortex regions after drinking HB, significant effects were found in the frontal, central, and occipital subregions, with *p*-values of 0.001, 0.000, and 0.031, respectively (Figure 2a). In contrast, a slight decrease in the delta spectral power was observed for all cortical subregions after drinking CB, showing no significant changes (*p* > 0.05) (Figure 2b). The ranking order of delta waves for both HB and CB was prefrontal, frontal, central, parietal, occipital, and temporal subregions. Significant changes in theta waves were observed in all cortical regions after drinking HB (*p* < 0.05) (Figure 2c). In contrast, no significant changes were observed after drinking CB (*p* > 0.05) (Figure 2d). Lagopoulos et al. (2009) reported a similar increase in theta power in the frontal, temporal, and central regions under meditation conditions representing calmness [13]. Although there was an increasing trend in all cortex subregions, no significant changes in alpha spectral powers were observed upon drinking HB (*p* > 0.05) (Figure 2e). However, after drinking CB, there was a significant decrease in the alpha spectral power in the prefrontal, frontal, central, temporal, and occipital cortex subregions with *p*-values of 0.039, 0.0001, 0.002, 0.007, and 0.024, respectively (Figure 2f). The effect of HB on beta waves was found only in the prefrontal and frontal subregions, showing a significant increase in μV^2^ (*p* < 0.05) (Figure 2g). Beta activity decreased significantly in the frontal and central cortical subregions upon the consumption of CB (*p* < 0.05) (Figure 2h).

### 3.2. Anterior-Posterior Analysis

Group analysis of anterior subregions (FP1, FP2, F4, Fz, F7, and F8) and posterior subregions (O1, O2, P3, P4, Pz, T5, and T6), without calculating the data from central electrodes (T3, T4, C3, C4, and Cz), was performed to compare the μV2 of delta, theta, alpha, and beta waves after drinking HB and CB (Figure 3). Comparing the subregions, all spectral powers in the anterior subregion showed higher levels than those in the posterior regions for both consumption types (hot and cold). The anterior subregion was denoted as the region of neurons for cognition, motivation, and execution, while organizing with neurons for sensory and somatosensory neurons in the posterior subregion [10]. Thus, it is noteworthy to use this anterior-posterior analysis to better compare relative changes across the cortical axis. Regarding delta waves, a significant increase was observed in both the anterior and posterior subregions after drinking HB (*p* = 0.012, *p* = 0.007, respectively) (Figure 3a). Although a slight decrease in delta power was observed in both regions after drinking CB, the changes were not statistically significant (*p* > 0.05) (Figure 3b). Similarly, a significant increase in theta waves was observed after drinking HB (*p* < 0.0001) in both subregions, despite no significant changes upon drinking CB (Figure 3c,d). The alpha and beta waves showed similar variation trends. In the anterior cortex, alpha and beta waves increased significantly after drinking HB, whereas they dropped significantly after drinking CB (*p* < 0.05) (Figure 3e–h). There were no significant effects on alpha and beta power in the posterior subregions after drinking HB (*p* > 0.05), but both showed a significant decrease after drinking CB (*p* < 0.05).

### 3.3. Hemisphere-Based Analysis

In hemisphere-based analysis, spectral data of electrodes from the left and right hemispheres before and after drinking HB and CB, respectively, were grouped and analyzed using a paired *t*-test (Figure 4). The spectral powers of both hemispheres are almost the same. In the case of spectral power upon consumption of beverages, the intake of HB significantly increased delta power in the left hemisphere and theta power in both hemispheres (*p* < 0.05) (Figure 4a,c). Intake of CB showed no significant changes in delta and theta (*p* > 0.05) (Figure 4b,d), but it declined significantly for alpha in both hemispheres and beta in the right hemisphere (*p* < 0.001) (Figure 4f,h).

### 3.4. Across Whole Cortex Analysis

Figure 5 displays the changes in EEG bands across the whole cortex pre- and post-intake of HB and CB, respectively, using a *t*-test. Both drinks demonstrated a similar ranking order of spectral power as follows: delta, alpha, beta, and theta. Changes of brain waves after drinking germinated wheat beverages (hot and cold) for four frequency bands in 19 EEG signals were determined using a *t*-test showing the statistical significance (*p* value) in Appendix A. Regarding channels, significant changes in delta waves were observed in the central (C4), frontal (F7), temporal (T4), and central (Cz) regions after the consumption of HB (*p* < 0.05), while there were no significant differences after drinking CB (*p* > 0.05). Consumption of HB showed significant changes in the delta, alpha, and theta bands (*p* < 0.05), but not the beta band (Figure 5a). There was a significant increase in the absolute power of delta (*p* < 0.0001), alpha (*p =* 0.029), and theta (*p* < 0.0001) after the consumption of HB. These outcomes are analogous with the alteration in brain activity after ingestion of a functionally active drink prepared with ginkgo and ginseng extract in a neurophysiological characterization study [14]. In contrast to drinking HB, the intake of CB had a significant effect only on the alpha and beta bands (*p* < 0.0001 and *p* < 0.01, respectively) (Figure 5b).

### 3.5. Topographic Analysis of the Whole Cortex

Figure 6 illustrates the absolute power differences for the delta, theta, alpha, and beta bands after intake of hot and cold beverages using a brain topographic graph. In this EEG topography, the values are color-coded and plotted with the lowest *p*-value in red to the highest *p*-value in blue. Figure 6a shows the significant changes in the EEG bands after drinking HB, except in alpha waves. Frontal, central, right temporal, and left occipital theta oscillations displayed significant changes (Figure 6a). The prefrontal and frontal beta regions showed highly significant changes after drinking HB (Figure 6a).

## 4. Discussion

In a study based on caffeine intake on EEG alpha power, caffeinated drinks suppressed the alpha wave more than the frontal, central, and occipital lobes and improved working memory [15]. Additionally, the attenuation of alpha power in the central and occipital cortical subregions is associated with better executive functioning and working memory [15,16]. According to the results of cortex-based analysis, intake of HB significantly increased most of the spectral powers in the frontal and central regions compared to other regions. Since the sense of smell and attention are associated functions in the frontal region, this enhancing phenomenon was similar to the effect of epigallocatechin gallate (EGCG) in tea, which is associated with both relaxation and alertness [17]. In conjunction with our previous findings, the content of volatiles and amino acids contained in HB and CB differed significantly (*p* < 0.05), showing a higher amount of most amino acids in CB compared to HB [2]. Supplementary analysis of correlated compounds with specific EEG bands on cortical subregions, using the values of compounds detected in HB and CB through high-performance liquid chromatography (HPLC) and electronic sensors as previously reported, revealed intriguing correlations with specific EEG bands across cortical subregions [2]. Notably, the prefrontal, frontal, temporal, parietal, and occipital cortex showed a negative correlation with most amino acids (histidine, serine, arginine, glycine, leucine, threonine, valine, proline, tyrosine, alanine, glutamic acid, and aspartic acid) and volatile compounds (2-methylheptane, 2-methyl-1-butanol, and methyl pentanoate), and positive correlation with other volatiles (methyl formate, 2-methylpentane, butane, 3-pentanol, and 1-hexen-3-ol) (Appendix A). However, central cortical beta spectral power depicted an inverse correlation trend (Appendix A). It is conceivable that the FFT absolute power of beta over the central region was relatively low after consuming both HB and CB (Figure 2). Consequently, the amino acid content of HB and CB appears to decrease beta bands in the central cortex.

In accordance with our results, higher delta and theta powers across the anterior and posterior regions were generally observed in meditators, implicating the condition more than the normal relaxed state, which reflects mindfulness or concentration [13]. Additionally, these greater changes of theta power in the anterior region than in the posterior region may be due to the fact that meditation specifically improves neural processing in the anterior cingulate cortex and limbic area, in which mental processing occur prominently [13]. Furthermore, higher theta and alpha in the frontal area were stated as a condition of attentiveness and mindfulness by decreasing sympathetic and increasing parasympathetic activity in a study based on meditation [13,18]. HB intake induces changes in brain activity that are analogous to calmness and relaxation associated with the meditation state. In contrast, the significant attenuation of the alpha and beta bands in the anterior-posterior region after drinking CB is consistent with the effect of caffeine on brain waves during the resting state in a previous study [19]. This reported that the spectral power of the alpha and beta bands decreased with the consumption of caffeinated drinks owing to the caffeine content, which stimulates attention and reduces fatigue. This is corroborated by results of previous findings that the compositions of amino acids and volatile compounds differed in HB and CB [2]. Conjointly, these components might affect EEG spectral changes over both anterior and posterior cortical subregions. In Appendix A, methyl formate, 2-methylpentane, butane, 3-pentanol, 1-hexen-3-ol, and 2-methylheptane and both EEG bands over anterior and posterior cortex were associated with positive correlation coefficients, whereas most amino acids were correlated with negative correlation coefficients. It is obvious that the compounds with positive coefficient were relatively greater in HB than in CB [2], accompanied by increased changes of EEG spectral powers in both subregions (Figure 3). In addition, those with negative coefficients were detected at higher concentrations in CB than in HB, which might be linked to decreased EEG spectral powers after drinking CB (Figure 3).

According to the concept of sensorial stimuli related to brain activity and motivational processes, the left hemisphere reflects an approach attitude, and the right hemisphere reflects a withdrawal attitude [5]. Comparing the two hemispheres, the right hemisphere exhibits a higher spectral power for all brain waves than the left hemisphere. It is possible that the right hemisphere responds more sensitively than the left hemisphere to taste perception, especially the umami taste [20]. This mechanism may be explained by the transport of umami ligands from taste bud cells into the central nervous system in the brain, which is related to emotions [20]. Supporting these results, it has already been reported in our previous study that an umami taste was detected in HB and CB beverages using an electronic tongue sensor [2]. In that study, umami taste was associated with levels of most amino acids (glycine, alanine, arginine, serine, threonine, lysine, leucine, valine, phenylalanine, tyrosine, glutamic acid, and aspartic acid) and volatile compounds (2-methylheptane, 2-methyl-1-butanol, methyl pentanoate, benzyl acetate, 2-octanol, and vinylbenzene) contained in HB and CB germinated wheat beverages. Appendix A showed the correlation of these components and changes of EEG bands in positive and negative correlations (Appendix A). Another point of view regarding the changes in EEG bands is the odor-related stimulation of brain waves. Several studies have suggested that food odors have neurophysiological stimuli which may affect EEG responsivity due to salient olfactory stimulation [21,22,23]. According to a previous study, inhalation of fragrance exhibited a considerable decrease in absolute alpha power in both the left and right hemispheres due to olfactory stimulation in the brain [24]. They also suggested that the increase or decrease in EEG absolute power might be related to fragrant-attributed chemicals in the samples. In our previous work, we reported the presence of volatile compounds associated with specific odors, such as burnt, fruity, and sweet odors, using an electronic nose sensor [2]. These statements were supported by a supplementary analysis of correlations between components and EEG spectra, as shown in Appendix A; volatile compounds (methyl formate, 2-methylpentane, butane, 3-pentanol, butylate, 1-hexen-3-ol, 2-methylheptane) were positively correlated with changes in delta, theta, alpha, and beta spectral power in both hemispheres. These compounds have been detected in HB and CB beverages and their possible aroma profiles have been described in our previous investigation [2].

Predominantly, delta waves indicate a sleeping state, whereas theta activity is associated with alertness, cognition, and active brain function [13]. This prominent increase in slow oscillation (delta and theta activity), in conjunction with deep relaxation, is caused by the anxiolytic potential of phenolic and flavonoid constituents in certain beverages, according to similar evidence of EEG activity after blackcurrant juice intake [25]. Similarly, the increase in theta waves after drinking the psychedelic tea ayahuasca was reported to reflect a deep relaxation state and reverie [26]. Furthermore, the combination of Mg, vitamin B, green tea, and *Rhodiola* enhances theta power across the entire brain, suggesting that the potential benefits of tea constituents enhance behavioral performance under stress conditions [27]. Generally, generation of alpha waves occurs during the state of relaxation, and alertness, whereas beta waves reflect the state of alertness, perception, and mental concentration [28]. The effect of HB consumption on alpha waves was prominent, with a significant increase after drinking (*p* = 0.029) (Figure 5a). These findings are similar to the changes in EEG alpha waves after drinking a refreshing drink due to the effect of the refreshing perception reported in the study of refreshing perception on brain oscillation performance [29]. These refreshing perceptions reflected in the increased alpha waves suggest that the intake of nutrients exerted an effect on arousal and cortical activation [28]. Additionally, it was noted that higher alpha waves correlated with reduced stress and comfortable conditions [7]. There is evidence that oral intake of γ-amino butyric acid (GABA), a major neurotransmitter, results in the inclination of alpha and beta waves in adults under stress [30,31]. The beneficial effect of gallic acid ameliorates the reduction in alpha power owing to its antioxidant and anti-inflammatory properties, thereby preventing oxidative stress damage in the brain [32]. These findings are consistent with the presence of higher amounts of gallic acid and GABA in roasted-germinated wheat, as described in our previous report [3].

The dropping down of alpha waves was noted as a sign of loss of consciousness, which happens in a person in a sleeping state [26]. Generally, caffeine suppresses alpha waves across the cortex as caffeine acts as a psychostimulant, resulting in cortical activation, excitability, and enhanced arousal [33]. It has been hypothesized that caffeine binds to adenosine receptors and decreases the inhibition of neuronal firing [15]. Moreover, improved working memory and attention have been linked to alpha wave suppression [15]. Changes in beta waves indicate awareness and alertness [34]. It can be speculated that the intake of HB contributes to relaxed and sedative effects, as well as anti-stress conditions, while CB intake is correlated with increased working memory and attention. According to the analysis of correlated compounds with specific EEG bands based on the previous study, alpha and beta bands showed the negative correlation of most amino acids (histidine, serine, arginine, glycine, leucine, threonine, valine, proline, tyrosine, alanine, glutamic acid, and aspartic acid) and volatile compounds (2-methylheptane, 2-methyl-1-butanol, and methyl pentanoate) (Appendix A). Furthermore, these compounds were found in larger concentrations in CB than in HB [2]. In contrast, the positively correlated compounds (methyl formate, 2-methylpentane, butane, 3-pentanol, and 1-hexene-3-ol) with EEG bands across the whole cortex were more abundant in HB than CB (Appendix A) [2]. Therefore, it is noteworthy that the decrease or increase in EEG spectral powers is associated with the constituents of beverages consumed.

Frontal midline theta activity may be related to memory processes and negatively correlated with anxiety levels [35,36,37]. Furthermore, increased frontal-midline theta was demonstrated as a parasympathetic response during meditation [13]. As beta power is highly associated with cognitive regulation, it reflects attention and decision-making [38]. The activation of the brain network in the increased beta powers revealed improved attention, working memory, and sensory-motor integration after refreshing drinks [28]. In accordance with these findings, HB intake may elicit a calming and relaxing response, as well as enhance concentration in consumers. Conversely, there were no significant changes in the EEG bands after drinking CB (Figure 6b). As shown in the topographic view, the intake of CB only slight changed beta waves in the left frontal cortex region.

Furthermore, the significant differences between components in HB and CB may exert varying cognitive effects, such as attention and relaxation. When comparing HB and CB, these differences in chemical composition are likely to have distinct impacts on post-consumption brain activity. In our previous study, we found that HB contained higher levels of volatile components, such as 2-metylpentane, butane and methyl formate, which contribute to its strong pungent and fruity flavors compared to CB [2]. It is worth noting that changes in EEG waves induced by odors have been observed in previous research. For instance, the increase in theta waves following exposure to various odors, such as birch tar, jasmine, lavender, lemon, and heliotropine, is associated with a reflection of drowsiness [21]. Consistent with these findings, the consumption of HB also resulted in an increase in theta waves, suggesting similar odor-induced effects on brain activity. In a previous study utilizing PCA analysis to compare the components of HB and CB, CB illustrated a clear separation due to its higher levels of amino acids, including glycine, histidine, leucine, phenylalanine, glutamic acid, aspartic acid, threonine, tyrosine, valine, lysine, proline, serine, and arginine [2]. Amino acids are building blocks of proteins, and they play crucial roles in various physiological processes, including neurotransmitter synthesis and brain function [39]. Of particular interest, supplementation of branched-chain amino acids (BCAAs) has been hypothesized as a means to accelerate the brain’s glutamate (GLU) disposal rate, thereby reducing the neurotoxic effects of excessive extracellular GLU, which can overstimulate neurons and lead to cell death (excitotoxicity) [40]. Our exploration of the differential effects of HB and CB on cognitive function and brain activity underscores the intricate relationship between beverage composition and neural responses. Moreover, the chemical disparities between HB and CB, notably the higher amino acid content in CB, emphasize the potential impact of nutritional components on brain function.

## 5. Limitations and Future Perspectives

The current research has focused only on the effects of hot and cold germinated wheat beverages on changes in brain waves using EEG studies. There are limitations, such as the low number of participants, and not assessing the participants’ consumption background, such as tea and coffee drinking habits. Additionally, a dose-dependent analysis and other physiological measures were not conducted in this study. Hence, further investigations, such as controlling for demographic information (age, consumption patterns, and use of drugs), are required to achieve a deeper understanding of the effects of HB and CB consumption.

## 6. Conclusions

In summary, the consumption of HB and CB elicits different neuromodulatory responses in the brain. In multiple approaches, the increasing trend of EEG spectral powers was observed after drinking HB, while demonstrating a decreasing trend after drinking CB. Taking into consideration the EEG oscillations, the delta and theta band powers increased significantly with the intake of HB, but the alpha and beta band powers decreased significantly with the intake of CB. This could be due to the presence of different levels of constituents, types of preparation, and temperatures of the beverages. Thus, there is an apparent correlation between increased calmness, relaxation, and consciousness, and HB consumption. The consumption of CB results in cortical activation, arousal-related responses, and enhanced attention. Our findings suggest that the consumption of germinated wheat beverages prepared using different methods exerts different effects on human brain function. Furthermore, cortical region-based analyses can provide a deeper understanding of cognitive performance in neuroimaging research.

## Figures and Tables

**Figure 1 foods-12-03493-f001:**
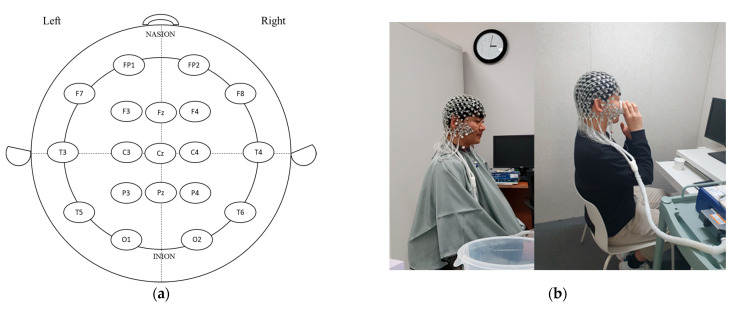
A representative diagram of electrode placement on the scalp using 10–20 international standard (**a**) and procedure of EEG measurement pre- and post-drinking of wheat beverage samples (**b**). FP: frontopolar (prefrontal); F: frontal; C: central; T: temporal; P: parietal; O: occipital; z (zero): sagittal midline.

**Figure 2 foods-12-03493-f002:**
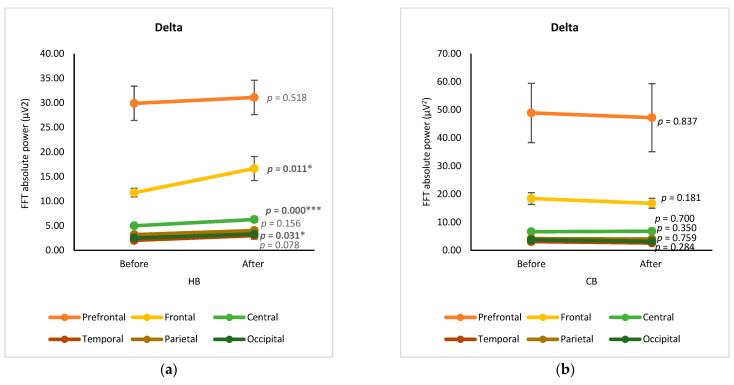
Changes of brain waves over cortical subregions. Changes of delta after drinking HB (**a**) and CB (**b**), theta after drinking HB (**c**) and CB (**d**), alpha after drinking HB (**e**) and CB (**f**), and beta after drinking HB (**g**), and CB (**h**), and illustration of cortex-based analysis (**i**). Each figure depicts mean average and error bars represent standard error of the mean (SEM) showing *p* values (*, *p* < 0.05; **, *p* < 0.001; ***, *p* < 0.0001) by paired *t*-test.

**Figure 3 foods-12-03493-f003:**
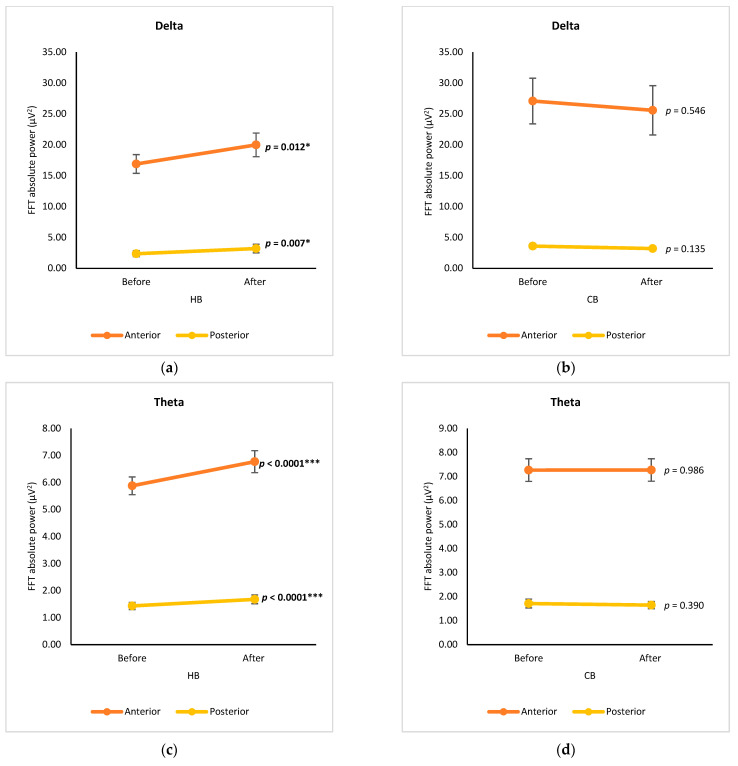
Changes of brain waves over anterior and posterior subregions. Changes of delta after drinking HB (**a**) and CB (**b**), theta after drinking HB (**c**) and CB (**d**), alpha after drinking HB (**e**) and CB (**f**), and beta after drinking HB (**g**), and CB (**h**), and illustration of anterior-posterior analysis (**i**). Each figure depicts mean average, and error bars represent standard error of the mean (SEM) showing *p* values (*, *p* < 0.05; **, *p* < 0.001; ***, *p* < 0.0001) by paired *t*-test.

**Figure 4 foods-12-03493-f004:**
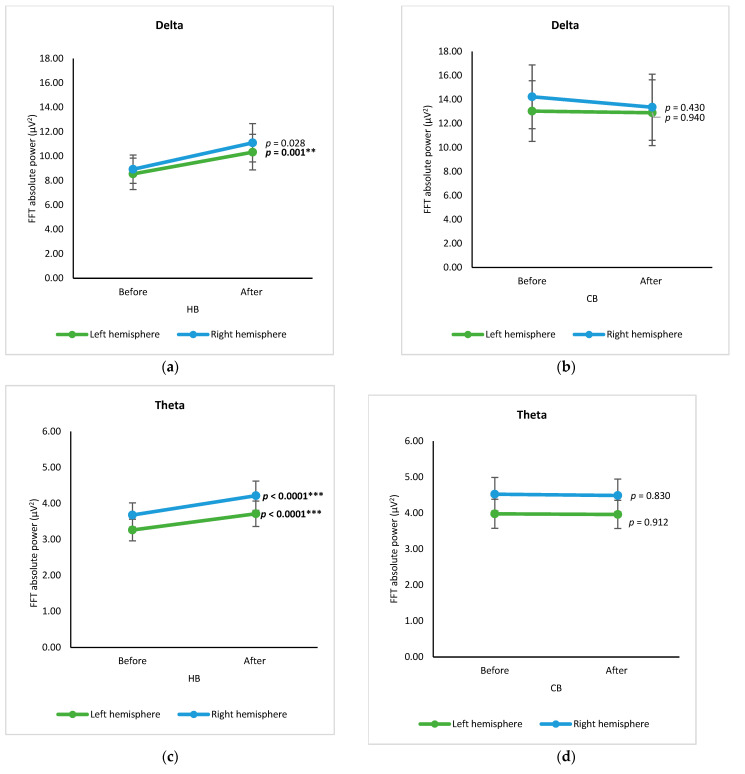
Changes of brain waves over the left and right hemisphere. Changes of delta after drinking HB (**a**) and CB (**b**), theta after drinking HB (**c**) and CB (**d**), alpha after drinking HB (**e**) and CB (**f**), and beta after drinking HB (**g**), and CB (**h**), and illustration of hemisphere-based analysis (**i**). Each figure depicts mean average, and error bars represent standard error of the mean (SEM) showing *p* values (**, *p* < 0.001; ***, *p* < 0.0001) by paired *t*-test.

**Figure 5 foods-12-03493-f005:**
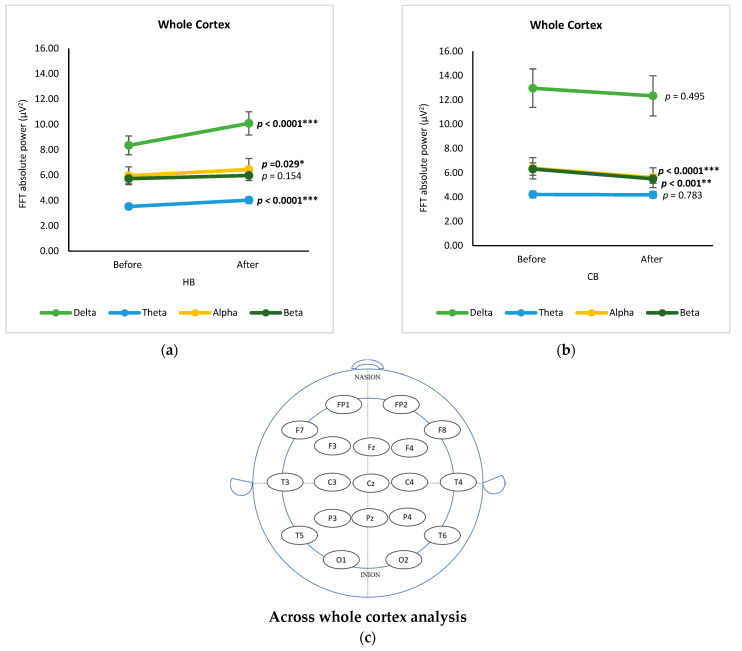
Changes of brain waves across whole cortex. Changes of delta, theta, alpha, and beta after drinking HB (**a**), and CB (**b**), and illustration of across whole cortex analysis (**c**). Each figure depicts mean average and error bars represent standard error of the mean (SEM) showing *p* values (*, *p* < 0.05; **, *p* < 0.001; ***, *p* < 0.0001) by paired *t*-test.

**Figure 6 foods-12-03493-f006:**
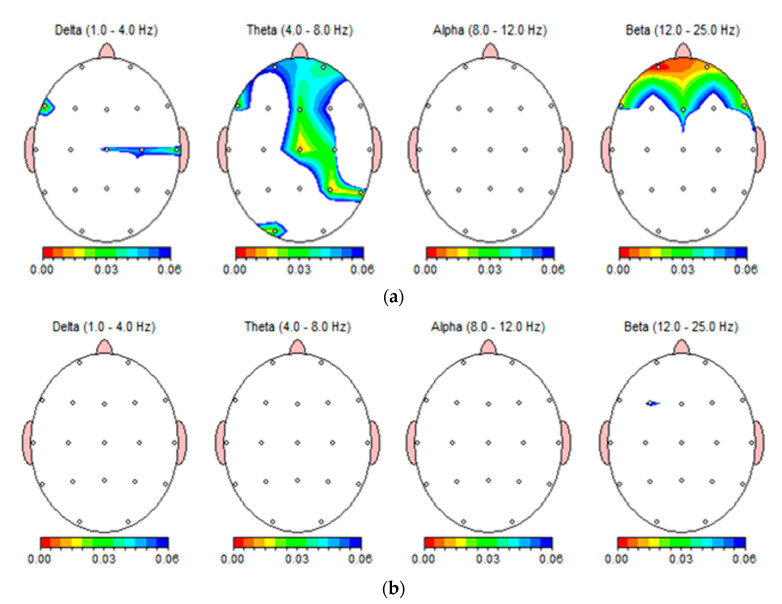
Brain topology of one subject expressing the absolute power differences in frequency bands: delta (1–4 Hz), theta (4–8 Hz), alpha (8–12 Hz), beta (12–25 Hz) after drinking HB (**a**), and CB (**b**).

## Data Availability

The data used to support the findings of this study can be made available by the corresponding author upon request.

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
