# Peer review of "Neuroimaging Approach: Effects of Hot and Cold Germinated Wheat Beverages on Electroencephalographic (EEG) Activity of the Human Brain"

_foods, 2023, doi:10.3390/foods12183493_

Round 1
Reviewer 1 Report
The manuscript entitled "Neuroimaging approach: Effects of hot and cold germinated-wheat beverages on electroencephalographic (EEG) activity of the human brain" is a good work done by the authors. Though the work appears to be interesting, there need more clarifications in several areas;
1. The abstract should clearly describe the purpose of study; what is the importance of neuroimaging and what is the significance of the hot/cold beverages on brain? A clear background section can be helpful to clarify this
2. What is the background of selecting hot and cold beverages? Authors should emphasize it in the introduction. Is there any difference in chemical composition of these two?
3. The methodology lacks a comparison of the composition of the two beverages. Authors should indicate the nutritional profile or chemical profile of these two beverages. Especially, the journal "foods" will be looking in to these properties of the beverages
4. Figure 2 and Figure 3 legends needs to be increased in size. Presently it is difficult to read (especailly the last image only). I think it is the "i" as other subfigures are names as "a - h"
5. There needs an extensive discussion of the obtained results with the chemical composition of the two beverages (which is to be carried out)
6. Minor typographic errors exist in the manuscript, which needs to be corrected
Author Response
Thank you for your comments. I attached response. Please see the file.

Reviewer 2 Report
The study employs neuroimaging techniques, specifically electroencephalography (EEG), to examine brain activity in consumers drinking hot and cold germinated wheat beverages (HB and CB, respectively). Four different analysis approaches were used to scrutinize brain wave changes—cortex-based, anterior-posterior-based, hemisphere-based, and whole cortex analyses, in addition to topographical assessments. The results showed that drinking HB led to an increase in delta and theta brain wave activity, which are generally associated with relaxation and calmness. Conversely, drinking CB led to a decrease in alpha and beta brain wave activity, generally linked to alertness and attention. The study suggests that HB could promote relaxation and mindfulness, while CB may enhance alertness and working memory. These neuroscientific methods could have broader applications in consumer research on beverage consumption. Overall, this research is very interesting to readers and food society, but all figures need to change from line plots to bar plots. Meanwhile, the statistical significance needs to be highlighted in the paper, especially in such a small participant group.
I can understand what the author wanna discuss. But a double Grammarly check would be better.
Reviewer 3 Report
This study shows that the consumption of wheat-based beverages, either hot (HB) or cold (CB) - germinated, elicits different neuromodulatory responses in the brain. Namely, the delta and theta band powers increased significantly with the intake of HB, while the alpha and beta band powers decreased significantly with the intake of CB. This was interpreted in terms that the intake of HB is probably related to relaxation, calmness, mindfulness and concentration, while the intake of CB is related to alertness, attention, and working memory. This is an interesting finding worthy for Foods.
A few minor observations are in order still:
Line 147: "10-20 international standard"... what? something seems to be missing here.
Define all acronyms used when they are first encountered (e.g. SEM)
Fig. S1 and the following: How were these chemical compounds identified and selected? Have you performed detailed chromatographic analyses of the beverages, in this study or a previous one? These chemical analyses should be detailed in the manuscript and discussed or the relevant reference(s) cited if it constitutes some of your past work.
Round 2
Reviewer 1 Report
No more comments